# The Impact of Hospital Volunteers’ Health Promotion Programs on Health Literacy and Quality of Life

**DOI:** 10.3390/healthcare13101134

**Published:** 2025-05-13

**Authors:** Chih-Hung Chen, Song-Seng Loke, Pi-Chi Han, Wei-Chuan Chen

**Affiliations:** 1College of Medicine, Chang Gung University, Taoyuan, Taoyuan City 33302, Taiwan; totoro631105@gmail.com; 2Division of Hepatogastroenterology, Department of Internal Medicine, Chang Gung Memorial Hospital-Kaohsiung Medical Center, Kaohsiung City 833401, Taiwan; 3Division of Geriatric Medicine, Department of Family Medicine, Kaohsiung Chang Gung Memorial Hospital, Kaohsiung City 833401, Taiwan; loke@cgmh.org.tw; 4Graduate Institute of Adult Education, National Normal Kaohsiung University, Kaohsiung City 80201, Taiwan; pichihan@gmail.com; 5Department of Medical Education and Research, Kaohsiung Veterans General Hospital, Kaohsiung City 813414, Taiwan; 6Department of Pharmacy and Master Program, Tajen University, Yanpu Township, Pingtung City 907101, Taiwan

**Keywords:** health literacy, life quality, volunteer, curriculum

## Abstract

**Background**: This study investigated whether a health literacy intervention program could improve the health literacy and quality of life among hospital volunteers. The study also explored the impact of health literacy on hospital volunteers’ health and psychological well-being. **Methods**: Overall, 35 hospital volunteers were recruited and divided into an experimental group (n = 22) and a control group (n = 13). The experimental group participated in an 8-week health literacy intervention program, which covered topics such as medication information, physiological and symptom-related vocabulary, and disease representation. The control group did not receive any intervention. A questionnaire survey was conducted to assess participants’ health literacy and quality of life before and after the intervention, and the comparison between two groups was statistically analyzed. **Results**: The experimental group showed significant improvements in multiple aspects of health literacy, particularly in medication information, physiology vocabulary, symptom vocabulary, and signs vocabulary (*p* < 0.05). In terms of quality of life, the experimental group demonstrated significant enhancements in psychological well-being and overall quality of life (*p* < 0.05). In contrast, the control group exhibited a downward trend in most health literacy dimensions with a significant decline in organ vocabulary (*p* < 0.05) and no significant changes in quality of life. **Conclusions**: The health literacy intervention program effectively improved hospital volunteers’ health literacy and quality of life with particularly notable effects on psychological well-being and the understanding of health-related professional terminology. By enhancing hospital volunteers’ health literacy and quality of life, healthcare organizations can foster more effective, sustainable, and satisfactory service quality.

## 1. Introduction

Hospital volunteers (“volunteers”) are critical in enhancing healthcare services, particularly by supporting patients and their families in improving health literacy. In addition to providing non-clinical assistance—such as greeting and guiding visitors, delivering meals, and helping with administrative tasks, volunteers often serve as essential intermediaries for improving patients’ understanding of health information. They assist patients in navigating complex healthcare systems, explaining hospital processes, and clarifying medical instructions, discharge guidelines, or preventive care measures. By facilitating communication between staff and patients, hospital volunteers help reduce confusion and empower patients to make informed decisions about their health [1]. Using volunteers to educate patients helps clinicians and improves compliance [2]. Research has underscored the importance of improving health literacy to enhance treatment outcomes, reduce hospital re-admissions, and promote self-care practices [3]. The literature supports that health literacy impacts health outcomes and quality of life. Volunteers can contribute to health promotion. Volunteers can identify gaps in understanding through interactions and connect patients with appropriate healthcare professionals for further guidance. Their contributions improve patient experiences and address health literacy disparities and critical health equity determinants [4].

However, quality of life (QoL) and health literacy significantly influence volunteer engagement and retention. QoL affects volunteers’ emotional capacity to perform their roles, while health literacy determines their ability to navigate and contribute effectively within healthcare environments. Understanding the interplay between these two factors is crucial for increasing volunteer participation and ensuring the sustainability of hospital volunteer programs. Our program uses experiential learning as a strategy. Experiential learning is a process in which students voluntarily participate in a series of activities and then analyze their experiences so that they can gain some knowledge and insights from them and apply these knowledge and insights to their own learning [5]. These insights are then applied to their own learning and future actions. In line with Lewin’s approach, our program follows a cyclical process that involves active participation, reflection, analysis, and reapplication. This cycle encourages learners to engage in experiences and critically evaluate and refine their approaches through continuous feedback.

Lewin’s model emphasizes that learning is not linear but involves an iterative process of action (concrete experience), observation (reflective observation), and planning (abstract conceptualization) for future experimentation (active experimentation). Each activity in our program is designed to initiate action and participation (concrete experience), followed by group discussions and self-reflection (reflective observation), where participants analyze their experiences and draw conclusions (abstract conceptualization). Finally, they plan new actions based on their insights, which they test in subsequent activities (active experimentation). In these teaching methods, we ensure that volunteers do not just gain knowledge but actively test and apply these insights in their daily lives and work, fostering a deeper, continuous learning process.

Health literacy, the ability to understand, access, and use health-related information, is vital for hospital volunteers. Volunteers with higher health literacy are better equipped to comprehend hospital policies, safety protocols, and patient care guidelines, enabling them to fulfill their duties effectively. Studies have shown that health literacy directly impacts volunteers’ confidence, ability, and retention rates, particularly in complex environments such as hospitals. Conversely, low health literacy can lead to misunderstandings, errors, and reduced confidence, hindering volunteer participation [3]. Volunteers involved in patient-facing roles, such as assisting with navigation or providing emotional support, require sufficient health literacy to meet the demands of their responsibilities. Targeted health literacy training has improved volunteer performance and retention rates. Such training enhances volunteers’ practical knowledge and boosts their confidence and motivation to continue volunteering [6]. Training programs focused on communication skills, medical terminology, and strategies for patient interaction can help volunteers feel more capable and engaged. Volunteers often act as patient advocates, helping patients understand medical instructions or navigate hospital services. Volunteers with higher health literacy are more effective at bridging communication gaps between patients and healthcare providers. Health literacy is critical to a volunteer’s ability to fulfill this advocacy role, especially when supporting vulnerable populations with limited health knowledge [4].

Quality of life, commonly defined as an individual’s perception of their physical, emotional, social, and psychological well-being, is another key factor influencing volunteer engagement. Volunteers with higher QoL tend to have greater energy, motivation, and emotional resilience, enabling them to participate in hospital volunteer work more actively and sustainably. Hospital volunteers often report high satisfaction and fulfillment, positively contributing to their QoL [7]. Volunteering has been shown to improve mental health, reduce depression, and strengthen social connections. In particular, hospital volunteers frequently develop a sense of purpose and belonging through their contributions, positively impacting their QoL [8]. Physical and emotional well-being are key components of QoL and critical determinants of volunteer engagement. Individuals with better physical and emotional health are more likely to sustain their involvement in hospital volunteer activities [9]. Additionally, the relationship between QoL and volunteering may vary with age [10]. Elderly hospital volunteers, for example, derive significant QoL improvements from volunteering, particularly in healthcare settings. However, age-related physical limitations may pose barriers to their participation. Insufficient health literacy can negatively impact volunteers’ QoL by leading to misunderstandings and reducing their confidence in performing tasks [11]. A meta-analysis of 23 studies revealed a moderate positive correlation (r = 0.35) between health literacy and QoL particularly in physical and mental health domains [12]. In short, QoL and health literacy are interdependent factors that jointly influence volunteer participation. Volunteers with good QoL and high health literacy are better positioned to maintain long-term involvement and contribute effectively to hospital operations. Research gaps remain despite the demonstrated importance of health literacy and QoL in hospital volunteer engagement. Current studies have primarily focused on the individual effects of health literacy or QoL on volunteer outcomes with limited attention to how targeted interventions, such as health literacy training, can simultaneously improve health literacy and QoL. In high-pressure environments like hospitals, factors such as experience, age, and education levels can affect the quality of volunteer services [13]. While the relationship between volunteering and QoL is well documented, the mechanisms by which improved health literacy influences QoL in hospital volunteering contexts remain unclear. Structured health literacy interventions may enhance health literacy and QoL, increasing volunteer engagement and retention. Therefore, this study aimed to address these gaps by exploring whether health literacy intervention programs can improve health literacy and QoL, providing actionable insights for hospital volunteer programs.

## 2. Materials and Methods

### 2.1. Study Design and Participants

This study employed a nonequivalent control group pre-test–post-test design. The participants were active hospital volunteers at a medical center recruited via convenience sampling. A total of 40 participants were enrolled and randomly assigned to either the experimental or the control group. The inclusion criteria were (1) willingness to sign an informed consent form and complete the intervention program and (2) ability to complete pre-test and post-test assessments. Recruitment was conducted through a public informational session, during which the principal investigator explained the study’s purpose and procedures. Participation was voluntary, and attendees decided whether to join the study.

### 2.2. Intervention

The experimental group participated in an 8-week “Holistic Care Model Health Promotion Program”, which consisted of weekly sessions lasting 2 h each. In contrast, the control group attended the same number of sessions for the same duration, but their activities were general gatherings that did not include health literacy-related content. Both interventions were conducted within the same time frame.

### 2.3. Data Collection

Data were collected using structured questionnaires. All participants completed a pre-test assessment before and a post-test assessment after the intervention concluded. The evaluation tools included a Health Literacy Scale and a Quality of Life (QoL) Scale. The study adhered to ethical research principles and received approval from the institution’s Human Research Ethics Committee.

### 2.4. Data Analysis

Appropriate statistical methods were used to evaluate the effects of the intervention. Descriptive and inferential statistics were applied to verify the impact of the health promotion program on the health literacy and QoL of hospital volunteers. This study uses *t*-test to compare the experimental group and control group, and paired-sample *t*-tests were used to compare pre- and post-intervention scores within groups.

## 3. Results

A total of 35 adults participated in this study with 22 in the experimental group and 13 in the control group. The baseline characteristics of the participants are summarized below (Table 1). The majority of participants were female. In the experimental group, 95.5% (21 participants) were female, while in the control group, 76.9% (10 participants) were female. The difference in gender distribution between the two groups was not statistically significant (*p* = 0.096). Most participants in the experimental group were aged 66–75 (63.6%), which was followed by those aged 56–65 (36.4%). In contrast, the age distribution in the control group was more evenly spread: 56–65 years (23.1%), 66–75 years (38.5%), and 76 years and older (38.5%). The difference in age distribution between the two groups was statistically significant (*p* = 0.007). In the experimental group, the majority (72.7%) had less than a high school education, whereas in the control group, 92.3% had less than a high school education. The difference in educational level distribution between the two groups was not statistically significant (*p* = 0.162).

In the experimental group, 50% of participants reported their health as “fair”, 36.4% as “average”, and 13.6% as “good”. In the control group, health status was more evenly distributed with 38.5% reporting “average” and 38.5% reporting “fair”. The difference in self-reported health status between the two groups was not statistically significant (*p* = 0.581). In the experimental group, 50% of participants reported having regular exercise habits, while in the control group, 69.2% reported having regular exercise habits. The difference between the two groups was insignificant (*p* = 0.267). In the experimental group, most participants reported either less than one year (40.9%) or more than three years (50%) of involvement in health promotion activities. In the control group, most participants reported less than one year of participation (46.2%). The difference between the two groups was not statistically significant (*p* = 0.391).

Aside from age distribution, there were no statistically significant differences in baseline characteristics between the experimental and control groups, indicating a relatively high degree of homogeneity between the two groups.

The Health Literacy Scale is designed based on the concept of health literacy defined by the Institute of Medicine of the United States through four different healthcare demand scenarios, including actual domestic health information, medical consultation conversations, medication instructions, and medical service documents. The Cronbach’s α of scale reliability is 0.95. The Quality of Life (QoL) Scale has 28 questions covering four major areas: physical health, psychology, social relations, environment, and general health status. The questionnaire uses a 5-point scale with higher scores indicating better quality of life. According to the preliminary test results of the Taiwan simplified version of the World Health Organization quality of life research group, Cronbach’s α of this questionnaire was as high as 0.91.

In Table 2, the experimental group demonstrated improvements across multiple dimensions following the health literacy intervention program. Paired sample *t*-test analyses revealed statistically significant improvements in four key areas: medication information (t = −3.177, *p* < 0.05), physiology vocabulary (t = −2.150, *p* < 0.05), symptom vocabulary (t = −2.353, *p* < 0.05), and signs vocabulary (t = −2.356, *p* < 0.05). While other dimensions did not reach statistical significance, they nonetheless exhibited positive trends of improvement.

These findings suggest that the health literacy intervention program effectively enhanced hospital volunteers’ understanding of health professional terminology and disease cognition. The most notable improvements were medication information comprehension, physiological vocabulary, symptom recognition, and disease representation.

As shown in Table 3, the experimental group also experienced improvements across various quality-of-life dimensions following the intervention. Paired sample *t*-test analyses indicated statistically significant progress in the psychological domain (t = −3.685, *p* < 0.05) and overall quality of life (t = −3.027, *p* < 0.05). Although other dimensions did not achieve statistical significance, they consistently demonstrated positive changes.

The results highlight the intervention program’s positive impact on hospital volunteers’ quality of life with particularly significant improvements in the psychological domain and overall life quality. These findings underscore the potential of targeted health literacy interventions to enhance professional understanding and contribute to personal well-being.

In Table 4, the control group demonstrated minimal changes between pre-and post-test measurements. The only statistically significant difference was observed in the “organ vocabulary” dimension (t = 2.285, *p* < 0.05), which showed a considerable decline. Across other dimensions, most scores exhibited a downward trend.

These findings suggest that in the absence of intervention, the control group’s health literacy failed to improve and demonstrated a decline across multiple dimensions. The significant reduction in organ health vocabulary comprehension is particularly noteworthy. This observation underscores the critical importance of targeted health literacy interventions.

Regarding the quality-of-life variables, no statistically significant differences were detected between pre-and post-test measurements for the control group.

The results highlight the potential deterioration of health literacy knowledge when no structured intervention is provided, emphasizing the need for continuous learning and supportive educational programs for healthcare volunteers.

## 4. Discussion

This study examined the effectiveness of a health literacy program in enhancing hospital volunteers’ health literacy and quality of life. The research framework was grounded in a comprehensive assessment of hospital volunteers’ health literacy status, utilizing empirical data analysis to explore the impact of health literacy [6,14]. Building upon this foundation, the study further investigated and developed an optimized intervention strategy and training program to elevate hospital volunteers’ health literacy levels [3].

The experimental group demonstrated significant improvements across multiple health literacy and quality of life domains, consistent with previous research, indicating the critical importance of health literacy education in enhancing volunteers’ health information comprehension and health-promoting behaviors [15]. In contrast, the control group showed no significant changes or declines in health literacy and quality of life indicators, emphasizing the pivotal role of targeted interventions in achieving positive outcomes for older volunteers in medical settings.

The results revealed that the health literacy intervention for hospital volunteers positively impacted their health knowledge and quality of life. Experimental group volunteers showed significant improvements in medication information, physiology vocabulary, symptom vocabulary, and signs vocabulary. The notable progress in medication information understanding is particularly significant, demonstrating the program’s effectiveness in helping volunteers better comprehend drug-related information. This is crucial for volunteers who provide basic health support and advice, enabling them to address patient and family queries about medication use effectively.

Given volunteers’ frequent interactions with patients and family members, enhanced health literacy strengthens their personal health management capabilities. It enables them to be more proactive in patient assistance and health education [3,6]. The control group’s decline in health literacy, especially organ vocabulary, potentially reflects knowledge retention challenges without ongoing health education support. These findings emphasize the importance of continuous health literacy education, particularly in improving volunteers’ application of health information [14].

Significant improvements in psychological and overall quality of life may be attributed to increased self-efficacy, confidence, and sense of purpose gained through health knowledge acquisition. Understanding and applying health information can reduce anxiety and enhance participants’ capacity to manage health effectively, resulting in improved quality-of-life outcomes. Although other quality-of-life domains did not reach statistical significance, they displayed positive trends, suggesting broader, more nuanced intervention impacts.

The health literacy intervention program demonstrates substantial clinical and economic value. By enhancing volunteers’ health literacy, they can more effectively support medical teams, alleviating frontline healthcare workers’ workload, particularly in medical consultation, basic health education, and patient support. Volunteers with foundational health knowledge can help reduce unnecessary medical expenses by preventing misunderstandings in medication use and health management.

However, the study has several limitations. First, the sample was restricted to volunteers from a single hospital, potentially limiting the result’s generalizability. Second, long-term intervention effects were not tracked. Moreover, health literacy measurements primarily relied on self-reported questionnaires, risking reporting bias. Future research should incorporate diverse measurement tools and consider experimental replication across regions and healthcare settings.

The significance of this research extends beyond filling existing theoretical gaps with primary emphasis on providing concrete, practical recommendations for healthcare institutions. By improving hospital volunteers’ health literacy, the study anticipated a dual benefit: enhancing individual volunteers’ quality of life and service effectiveness while potentially improving overall healthcare service quality [16]. The ultimate objective was to construct a comprehensive intervention program exploring the interrelationships between health literacy, quality of life, and voluntary service [4,17].

The findings underscore the effectiveness of the health literacy intervention program in improving hospital volunteers’ health literacy and quality of life.

## 5. Conclusions

This study demonstrates that health literacy intervention can significantly improve hospital volunteers’ health literacy and quality of life. Enhanced service quality, particularly in medication information and symptom identification, equips volunteers with essential skills for navigating medical environments. The positive impacts on psychological health and overall quality of life highlight the broader benefits of such programs. These findings emphasize the importance of integrating health literacy courses into volunteer training, improving knowledge and skills, and enhancing personal well-being and satisfaction. Based on the study’s findings, the effectiveness of implementing an intervention strategy depends on the hospital’s developmental management approach. The results improve hospital volunteers’ health literacy and quality of life, suggesting adding health literacy courses to the volunteers in on-the-job training in the hospital. Integrating the volunteer training program into future development programs may answer the question of the effectiveness of this strategy.

## Figures and Tables

**Table 1 healthcare-13-01134-t001:** Current status of participants.

Item		Experimental		Control		*p*-Value
		N	%	N	%	
Gender	Male	1	4.5%	3	23.1%	0.096
	Female	21	95.5%	10	76.9%	
Age	56–65	8	36.4%	3	23.1%	0.007
	66–75	14	63.6%	5	38.5%	
	Above 76	0	0%	5	38.5%	
Health Status	Bad	0	0%	1	7.7%	0.581
	Average	8	36.4%	5	38.5%	
	Fair	11	50%	5	38.5%	
	Good	3	13.6%	2	15.4%	
Exercise	Yes	11	50%	9	69.2%	0.267
	No	11	50%	4	30.8%	

N = 35.

**Table 2 healthcare-13-01134-t002:** Effectiveness of a health literacy program on hospital volunteers: experimental group.

Variables	Pre-Test	Post-Test	t
	M (SD)	M (SD)	
Health Education Pamphlet	8.68 (1.04)	8.95 (1.13)	−1.299
Physician–Patient Dialogue	11.68 (0.57)	11.55 (0.6)	1
Medication Information	15.41 (1.92)	16.32 (0.89)	−3.177 *
Medical Service System	9.55 (1.37)	9.32 (1.04)	0.642
Medicine Vocabulary	20.77 (6.09)	24.59 (7.06)	−2.003
Severe Illness Vocabulary	27.73 (8.11)	31.23 (8.59)	−1.475
General Diseases	27.23 (8.34)	31.45 (8.62)	−1.714
Organ Vocabulary	16.77 (6.31)	20.27 (6.34)	−1.977
Physiology Vocabulary	18.05 (6.17)	21.68 (6.47)	−2.150 *
Examination Process Explanation	19.68 (6.17)	23.14 (7.09)	−2.033
Treatment Procedures	13.95 (4.51)	16.45 (5.3)	−1.879
Symptom Vocabulary	17.82 (5.68)	21.77 (6.6)	−2.353 *
Signs Vocabulary	12.36 (4.44)	15.14 (4.59)	−2.356 *

N = 22, * *p* < 0.05.

**Table 3 healthcare-13-01134-t003:** Effectiveness of a quality of life on hospital volunteers: experimental group.

Domain	Pre-Test	Post-Test	t
	M (SD)	M (SD)	
Physical health	25.14 (2.83)	26.18 (3.08)	−1.771
Psychological	19.59 (2.82)	21.23 (2.43)	−3.685 *
Social relations	13.68 (2.17)	14.41 (2.15)	−2.012
Environmental	31.05 (4.15)	32.77 (3.28)	−1.934
Overall	95.95 (11.54)	101.45 (9.73)	−3.027 *

N = 22, * *p* < 0.05.

**Table 4 healthcare-13-01134-t004:** Effectiveness of a health literacy program on hospital volunteers: control group.

Variables	Pre-Test	Post-Test	t
	M (SD)	M (SD)	
Health Education Pamphlet	7.92 (1.32)	7.54 (1.76)	0.789
Physician–Patient Dialogue	10.92 (1.66)	10.77 (1.01)	0.379
Medication Information	14.46 (1.61)	14.92 (1.44)	−0.945
Medical Service System	8.15 (1.99)	8.69 (1.44)	−1.047
Medicine Vocabulary	26.08 (6.5)	26.46 (8.48)	−0.065
Severe Illness Vocabulary	33.85 (9.85)	31.54 (13.49)	0.644
General Diseases	32.92 (10.66)	30.62 (13.87)	0.654
Organ Vocabulary	21.15 (7.06)	17.38 (8.39)	2.285 *
Physiology Vocabulary	23.15 (6.56)	19.46 (8.61)	1.697
Examination Process Explanation	23.77 (7.47)	21.15 (9.19)	1.107
Treatment Procedures	16.46 (4.77)	14.54 (7.43)	1.065
Symptom Vocabulary	22.69 (6.85)	20.62 (9.86)	0.732
Signs Vocabulary	16.31 (6.06)	14.62 (7.56)	0.688

N = 13, * *p* < 0.05.

## Data Availability

Data are not shared due to privacy and ethical restrictions.

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
