# Peer review of "The Impact of Hospital Volunteers’ Health Promotion Programs on Health Literacy and Quality of Life"

_healthcare, 2025, doi:10.3390/healthcare13101134_

Round 1
Reviewer 1 Report
Comments and Suggestions for Authors
The aim of the study is to show that a health literacy intervention programme improves the health literacy and quality of life of hospital volunteers. This relationship seems obvious: if there is a programme to improve the health literacy of participants, then their health literacy must be improved, unless the programme isn't working properly. If the role of a volunteer is also to improve patient understanding, and volunteers do not have the necessary information, then their well-being must be at a lower level. In the introductory chapter, it would be useful to argue why the need to train volunteers was doubtful, whether the hospital environment (management) or local authorities or general feeling was against the implementation of such an intervention programme. Without this kind of background information, the importance of the research is not convincing.
Some background literature would also be advisable to highlight the scientific and practical importance of the study. The actual literature review is about the relationship between the health literacy of volunteers and their ability to perform their tasks. However, the research focuses on the effects of a specific training programme, i.e. it evaluates a single information programme. In my opinion, the research would have been more relevant if the research question had focused on the effectiveness of different information programmes, or the effectiveness of a specific programme compared to other types of programmes known from the literature. This means that the literature review, based on the available information, should have discussed the advantages and disadvantages of different training programmes and the theoretical considerations behind these programmes, they should have presented the introduced and evaluated programme, what was the concept, what was the concrete aim of the implementation, and based on the pre- and post-test results the authors could have evaluated the effectiveness of the programme.
In the methodological chapter, a more concrete description of the statistical analysis is advisable: 'appropriate statistical methods', as the description of the method used is not an adequate description. Although they mention that they used the Health Literacy Scale and the QoL Scale, I miss some more information about the variables involved in the analyses: what kind of variables they used and what they measured with these variables. If there were any theoretical considerations when selecting the variables, these considerations should have been reviewed in the introductory chapter. For example, what were they measuring with the health status, physical activity of the participants, and why was this important from the point of view of the research topic.
The statistical analyses are presented correctly, but the result is not very surprising: the health literacy of those who participated in the programme improved at the end of the study.
The results on the status of the participants give a certain overview of the sample, but they should have been used as explanatory variables in the statistical analyses. (The impact of the programme is different for men and women, for younger and older, etc.) A surprising result is that in the case of the control group, the post-test showed a significant decline in the case of "organ vocabulary". I miss an explanation here.
The conclusion chapter reflects the empirical results of the study well. The authors should highlight the novelty of their research, in what way their results are similar to previous other studies, and what is different in their results from those reviewed in the literature.
It would have been more useful to focus on the programme itself and how the research findings were implemented when developing an intervention strategy. In the future they should reveal the effectiveness of the strategy developed.
References are appropriate, but a few more would have been welcome.
Author Response
Dear Reviewer 1,
Comment: The aim of the study is to show that a health literacy intervention programme improves the health literacy and quality of life of hospital volunteers. This relationship seems obvious: if there is a programme to improve the health literacy of participants, then their health literacy must be improved, unless the programme isn't working properly. If the role of a volunteer is also to improve patient understanding, and volunteers do not have the necessary information, then their well-being must be at a lower level. In the introductory chapter, it would be useful to argue why the need to train volunteers was doubtful, whether the hospital environment (management) or local authorities or general feeling was against the implementation of such an intervention programme. Without this kind of background information, the importance of the research is not convincing.
Author’s reply:
Thank you for the suggestion. Volunteers in hospitals are always composed of elderly people in Taiwan. As the article hospital volunteers are critical in enhancing healthcare services, although hospitals have hospital volunteer programs. This study uses quantitative research to prove that through health literacy intervention program can significantly affect to improving quality of life and health literacy of volunteer.
- Resilience of volunteer through holistic healthcare training, which contains spiritual learning lessons can improve resilience. Our programme actually includes this training and not shown in the article.
- Elderly volunteers need more attention for their health, and hospital volunteers need to serve many patients. Therefore, if the health literacy and quality of life of volunteers can be effectively improved through courses, we believe that they can not only reduce their own chances of getting sick but also give back to other patients or families they help. We believe that this plan is both accessible and feasible.
- Retention: Future research will further verify the improvement of volunteer participation and retention rate because retention and participation will affect service quality.
Comment: Some background literature would also be advisable to highlight the scientific and practical importance of the study. The actual literature review is about the relationship between the health literacy of volunteers and their ability to perform their tasks. However, the research focuses on the effects of a specific training programme, i.e. it evaluates a single information programme. In my opinion, the research would have been more relevant if the research question had focused on the effectiveness of different information programmes, or the effectiveness of a specific programme compared to other types of programmes known from the literature. This means that the literature review, based on the available information, should have discussed the advantages and disadvantages of different training programmes and the theoretical considerations behind these programmes, they should have presented the introduced and evaluated programme, what was the concept, what was the concrete aim of the implementation, and based on the pre- and post-test results the authors could have evaluated the effectiveness of the programme.
Author’s reply:
Thank you for the comment. This health literacy intervention program aims to improve hospital volunteers' health knowledge and quality of life. Through this project, we can give volunteers more comprehensive health knowledge to provide more effective support and guidance when assisting patients while reducing their chances of getting sick and thus improving their quality of life. Compared with other programs in the literature, this program differs from previous narrative courses. Examples are mainly based on localized situational cases. The course design of each interactive health course combines functional, interactive, and critical characteristics. The course objectives are pre-set in the lives and environments faced by these elderly volunteers. Combined with group activities and discussions, the participants are guided to reflect on their own needs first and then combined with group discussions and group learning in the classroom to improve learning efficiency and breadth; through the development of appropriate teaching materials and programs, the reality and concerns of the daily lives of older people are connected. This research focuses on the effects of a specific training program to evaluate the effectiveness of a health literacy intervention program. We can discuss the advantages and disadvantages of different training programs and the theoretical considerations behind these programs in the future.
Comment: In the methodological chapter, a more concrete description of the statistical analysis is advisable: 'appropriate statistical methods', as the description of the method used is not an adequate description. Although they mention that they used the Health Literacy Scale and the QoL Scale, I miss some more information about the variables involved in the analyses: what kind of variables they used and what they measured with these variables. If there were any theoretical considerations when selecting the variables, these considerations should have been reviewed in the introductory chapter. For example, what were they measuring with the health status, physical activity of the participants, and why was this important from the point of view of the research topic.
Author’s reply:
Statistic methods:
We updated the statistical analysis methods in “data analysis” section (line147-149).
Comment: The statistical analyses are presented correctly, but the result is not very surprising: the health literacy of those who participated in the programme improved at the end of the study.
Author’s reply:
Certainly, it is logically hypothesized that volunteers who participated in the programme can improved at the end of the study. The purpose of this program is to establish the fundamental skills of volunteers to cope with patients. Therefore, the first consideration of assessing straining outcome of Health literacy and Quality of life improvement is to compare quantitatively.
Our program is about:
- Resilience of volunteers through holistic healthcare training, which contains spiritual learning lessons, can improve resilience. Our programme actually includes this training and not shown in the article.
- This is exactly the research result we are concerned about. Because we used a teaching method that is different from the previous narrative teaching. This study aims to compare the differences between different groups after the course intervention. Therefore, the design adopts a quasi-experimental method. The first focus is on the changes within the group before and after the course intervention, and the second is the difference in the final results between the groups. Therefore, the paired sample t-test method was mainly adopted because this method directly compares the data of each participant at two time points, which is crucial for understanding the effect of the course intervention. Finally, the course presentation will be helpful to the volunteer team of our institute.
Comment: The results on the status of the participants give a certain overview of the sample, but they should have been used as explanatory variables in the statistical analyses. (The impact of the programme is different for men and women, for younger and older, etc.) A surprising result is that in the case of the control group, the post-test showed a significant decline in the case of "organ vocabulary". I miss an explanation here.
Author’s reply:
Probably, the professional medical term may confuse the participants. This result may be further interpretated by the qualitative study to understand the reason.
Comment: The conclusion chapter reflects the empirical results of the study well. The authors should highlight the novelty of their research, in what way their results are similar to previous other studies, and what is different in their results from those reviewed in the literature.
Author’s reply:
Maybe, the results of similarity to the Western countries imply that the training program is essential to hospital volunteers. Therefore, implementation of the training program is important. To our knowledge, this is the first study of the training program of hospital volunteers in Taiwan. This study provides an important evidence.
Comment: It would have been more useful to focus on the programme itself and how the research findings were implemented when developing an intervention strategy. In the future they should reveal the effectiveness of the strategy developed.
Author’s reply:
Thanks to the reviewer for the comment. In the conclusion of manuscript, it was shown that the positive impacts on psychological health and overall quality of life highlight the broader benefits of such programs. These findings emphasize the importance of integrating health literacy courses into volunteer training, improving knowledge and skills, and enhancing personal well-being and satisfaction. Given the outcome in this study , the effectiveness developing an intervention strategy depends on the strategy of hospital developing management. The volunteers training program integrating into faulty development program in the future the way may answer the question of the effectiveness of this strategy. ( line301-304 )
Comment: References are appropriate, but a few more would have been welcome.
Author’s reply:
Thanks for the suggestion, we have added more literature.
Reviewer 2 Report
Comments and Suggestions for Authors
Primary Weaknesses (Must Be Addressed Before Publication)
-
Lack of Power Analysis
There is no explanation of how the sample size (n = 35) was determined. The absence of a power analysis undermines the statistical strength of the findings and raises concerns regarding the generalizability of the results. -
Weak Study Design (Non-Equivalent Groups)
Although the study claims random assignment, the age distribution between the two groups was significantly different (p = 0.007), indicating potential selection bias. This inconsistency questions the internal validity of the intervention. -
Insufficient Statistical Analysis
The study relies solely on paired t-tests without adjusting for baseline differences between the groups. No use of ANCOVA or other methods to control for covariates or baseline imbalances, which is critical in quasi-experimental designs. -
Exclusive Use of Self-Reported Measures
The study relies entirely on self-administered questionnaires for both health literacy and quality of life, without incorporating objective assessment tools. This increases the risk of response bias and limits the reliability of the findings. -
Lack of Psychometric Information
The manuscript does not report the validity or reliability of the instruments used to measure health literacy and quality of life. This omission reduces the credibility and replicability of the measurement approach. -
No Assessment of Long-Term Effect
The study does not include any follow-up evaluation to assess the sustainability of the observed improvements post-intervention. This limits conclusions about long-term effectiveness. -
No Between-Group Analysis of Change
The analysis focuses only on within-group comparisons (pre- vs. post-test) and omits any statistical comparison of change scores between the intervention and control groups (e.g., interaction effect), which is essential in experimental or quasi-experimental research.Secondary Weaknesses (Suggested Improvements)
-
Title Does Not Reflect the Presence of a Control Group
It is recommended to revise the title to better reflect the study's comparative nature. For example:
"Effectiveness of a Health Literacy Program on Hospital Volunteers: A Controlled Trial" -
Non-Standard Presentation of Tables
The formatting of the tables is inconsistent. For example, the asterisk indicating statistical significance (*p < .05) should be clearly and consistently presented in all relevant cells and table footnotes. -
Limited Geographical and Demographic Representation
The sample is limited to a single hospital in Taiwan, which restricts the external validity and generalizability of the findings to broader populations. -
Lack of Theoretical Basis for the Training Program
The paper does not explain whether the intervention program was based on an established theoretical model (e.g., Bandura's Social Learning Theory or the PRECEDE-PROCEED Model), which is critical for ensuring scientific rigor in intervention design. -
Outdated References
While the manuscript includes some relevant citations, many of the references are dated before 2012. Updating the literature with more recent studies would strengthen the theoretical foundation and relevance of the research.
-
The manuscript is generally understandable; however, the quality of English can be improved to enhance clarity and readability. Several sentences contain grammatical errors, awkward phrasing, and redundancy, which affect the professional tone of the paper. I recommend a thorough revision by a native English speaker or a professional scientific editor to improve sentence structure, academic style, and consistency across the manuscript.
Author Response
Dear Reviewer 2,
Primary Weaknesses (Must Be Addressed Before Publication)
- Lack of Power Analysis
Comment: There is no explanation of how the sample size (n = 35) was determined. The absence of a power analysis undermines the statistical strength of the findings and raises concerns regarding the generalizability of the results.
Author’s reply:
- This is a pilot study
- The 35 participants are retired people and enrollment of volunteers is not easy in Taiwan. The role of hospital volunteers was only as assistants in helping in administration. The fundamental training for them is also thought difficult for them. it is not easy for them to participate in the pre- and post-tests, especially is difficult for the elderly participants.
Weak Study Design (Non-Equivalent Groups)
Comment: Although the study claims random assignment, the age distribution between the two groups was significantly different (p = 0.007), indicating potential selection bias. This inconsistency questions the internal validity of the intervention.
Author’s reply:
It is not easy for the elderly participants to join the program. Besides, there were no significant differences in other background conditions. Although there is a significant difference in age, the bottom line also meets the needs of this study.
- Insufficient Statistical Analysis
Comment: The study relies solely on paired t-tests without adjusting for baseline differences between the groups. No use of ANCOVA or other methods to control for covariates or baseline imbalances, which is critical in quasi-experimental designs.
Author’s reply:
This study aims to compare the differences between pre-test and post-test groups after the course intervention. Therefore, the quasi-experimental method is designed to focus first on the changes within the group before and after the course intervention, and then on the differences in the final results between the groups. Therefore, the paired sample t-test method was mainly adopted because this method directly compares the data of each participant at two time points, which is crucial for understanding the effect of course intervention.
- Exclusive Use of Self-Reported Measures
Comment: The study relies entirely on self-administered questionnaires for both health literacy and quality of life, without incorporating objective assessment tools. This increases the risk of response bias and limits the reliability of the findings.
Author’s reply:
The reliability and validity data of the questionnaire have been supplemented in the manuscript, Correction part shown in the line 180-189.
- Lack of Psychometric Information
Comment: The manuscript does not report the validity or reliability of the instruments used to measure health literacy and quality of life. This omission reduces the credibility and replicability of the measurement approach.
Author’s reply:
The reliability and validity data of the questionnaire have been supplemented in the manuscript, please see the line 180-189.
- No Assessment of Long-Term Effect
Comment: The study does not include any follow-up evaluation to assess the sustainability of the observed improvements post-intervention. This limits conclusions about long-term effectiveness.
Author’s reply:
This study is a pilot study and we will continue to track the effectiveness of the course of program in the future.
- No Between-Group Analysis of Change
Comment: The analysis focuses only on within-group comparisons (pre- vs. post-test) and omits any statistical comparison of change scores between the intervention and control groups (e.g., interaction effect), which is essential in experimental or quasi-experimental research.
Author’s reply:
This study aims to compare the differences between different groups after the course intervention. Therefore, the quasi-experimental method is designed to focus first on the changes within the group before and after the course intervention, and then on the differences in the final results between the groups. Therefore, the paired sample t-test method was mainly adopted because this method directly compares the data of each participant at two time points, which is crucial for understanding the effect of the course intervention. Furthermore, given our sample size and the simplicity and interpretability of our analyses, we believe that paired-sample t tests more directly answer our research questions regarding within-group variation.
Secondary Weaknesses (Suggested Improvements)
- Title Does Not Reflect the Presence of a Control Group
Comment: It is recommended to revise the title to better reflect the study's comparative nature. For example:
"Effectiveness of a Health Literacy Program on Hospital Volunteers: A Controlled Trial"
Author’s reply:
Thanks you, the title has been revised (line200, 212, 228).
- Non-Standard Presentation of Tables
Comment: The formatting of the tables is inconsistent. For example, the asterisk indicating statistical significance (*p < .05) should be clearly and consistently presented in all relevant cells and table footnotes.
Author’s reply:
We modify the format of the tables. Thank you.
- Limited Geographical and Demographic Representation
Comment: The sample is limited to a single hospital in Taiwan, which restricts the external validity and generalizability of the findings to broader populations.
Author’s reply:
Thank you. This study is a pilot study and we look forward to expanding to other hospitals for ongoing research in the future.
- Lack of Theoretical Basis for the Training Program
Comment: The paper does not explain whether the intervention program was based on an established theoretical model (e.g., Bandura's Social Learning Theory or the PRECEDE-PROCEED Model), which is critical for ensuring scientific rigor in intervention design.
Author’s reply:
The theoretical basis used in this project has been added to lines 64-68.
- Outdated References
Comment: While the manuscript includes some relevant citations, many of the references are dated before 2012. Updating the literature with more recent studies would strengthen the theoretical foundation and relevance of the research.
Author’s reply:
We added more recent references into this study for strengthen the relevance of the research (line50-55, 64-68, 104-106).
Round 2
Reviewer 1 Report
Comments and Suggestions for Authors
The reviewers' comments have been satisfactorily addressed by the authors. The additions have clarified the research question and methodology. It would be worthwhile to include some of the responses to the comments in the article, such as sentences describing the aim of the health literacy programme, or describing the specificity of the programme ('we used a teaching method that is different from the previous narrative teaching') and its significance ('this is the first study on the training programme of hospital volunteers in Taiwan').
Author Response
Dear Reviewer 1,
Comments1
The reviewers' comments have been satisfactorily addressed by the authors. The additions have clarified the research question and methodology. It would be worthwhile to include some of the responses to the comments in the article, such as sentences describing the aim of the health literacy programme, or describing the specificity of the programme ('we used a teaching method that is different from the previous narrative teaching') and its significance ('this is the first study on the training programme of hospital volunteers in Taiwan').
Reply1
Thank the suggestions and we have change sentences describing the aim, as “this is the first study on the teaching methods of hospital volunteers in Taiwan.
Reviewer 2 Report
Comments and Suggestions for Authors
Comments and Suggestions for Authors
Thank you for the revised manuscript. The improvements made to the introduction and discussion sections are notable, especially the added explanations regarding experiential learning and the use of updated references. The following points are offered to further enhance the clarity, academic rigor, and consistency of your manuscript:
-
Clarification of Methodology: You have added helpful detail about the tools used, including Cronbach’s alpha values. However, the explanation of statistical tests (p. 4, line 147–149) needs clarification and correction. The phrase "Using t-test analyses to be revealed statistically..." is grammatically incorrect. Please revise to state clearly that paired-sample t-tests were used to compare pre- and post-intervention scores within groups.
-
Theoretical Integration: The added reference to experiential learning is appreciated, but the paragraph introducing it (p. 2, lines 63–68) could be better connected to the study’s context. Consider explaining how experiential learning principles guided the design of the intervention program, rather than simply describing the theory.
-
Language and Style Consistency: Several sentences across the manuscript (especially the conclusion and added content in the discussion) would benefit from clearer academic phrasing. For example, the final sentence: "Given the outcome in this study , the effectiveness developing an intervention strategy depends on the strategy of hospital developing management..." needs revision for clarity and grammar.
-
Formatting and Spacing: There are minor typographic issues (e.g., inconsistent spacing before punctuation, especially commas in citations and tables). These should be corrected for professional presentation.
-
Conclusion Expansion: The conclusion successfully summarizes the findings, but you might enhance it by briefly reaffirming how these results contribute to broader practice or policy regarding volunteer training in hospitals.
-
References: The updated references are relevant and strengthen the study. Please ensure consistency in formatting according to journal guidelines, especially for multi-author listings and DOIs if required.
Author Response
Dear Reviewer 2,
Comments 2
Thank you for the revised manuscript. The improvements made to the introduction and discussion sections are notable, especially the added explanations regarding experiential learning and the use of updated references. The following points are offered to further enhance the clarity, academic rigor, and consistency of your manuscript:
1. Clarification of Methodology: You have added helpful detail about the tools used, including Cronbach’s alpha values. However, the explanation of statistical tests (p. 4, line 147–149) needs clarification and correction. The phrase "Using t-test analyses to be revealed statistically..." is grammatically incorrect. Please revise to state clearly that paired-sample t-tests were used to compare pre- and post-intervention scores within groups.
Reply: Thank you. We modify the statement for clearly the phrase. (p. 4, line 147–149)
2. Theoretical Integration: The added reference to experiential learning is appreciated, but the paragraph introducing it (p. 2, lines 63–68) could be better connected to the study’s context. Consider explaining how experiential learning principles guided the design of the intervention program, rather than simply describing the theory.
Reply: We updated experiential learning principles guided into the statement. Please see line 67-81.
3.Language and Style Consistency: Several sentences across the manuscript (especially the conclusion and added content in the discussion) would benefit from clearer academic phrasing. For example, the final sentence: "Given the outcome in this study , the effectiveness developing an intervention strategy depends on the strategy of hospital developing management..." needs revision for clarity and grammar.
Reply: Thank you. We modify the statement. Please see line 301-311.
4.Formatting and Spacing: There are minor typographic issues (e.g., inconsistent spacing before punctuation, especially commas in citations and tables). These should be corrected for professional presentation.
Reply: Revised format. Thank you.
5.Conclusion Expansion: The conclusion successfully summarizes the findings, but you might enhance it by briefly reaffirming how these results contribute to broader practice or policy regarding volunteer training in hospitals.
Reply: We added the suggestion into conclusion, please see line 309-311.
6.References: The updated references are relevant and strengthen the study. Please ensure consistency in formatting according to journal guidelines, especially for multi-author listings and DOIs if required.
Reply: We do our best to find the DOIs, but three references (no. 1,5,10) do not have DOI. Revised format. Thank you.